# *Xanthomonas citri* subsp. *citri* requires a genus-specific outer membrane protein and TolB to coordinate cell membrane integrity and virulence

Leilei Wu,[1] Xueyan Zhu,[1] Hang Long,[1] Xinyu Huang,[1] Shuying Zhu,[1] Qi Xue,[1] Ziao Li,[1] Huasong Zou[1]

**ABSTRACT**  *Xanthomonas citri* subsp. *citri* (*Xcc*) possesses a *Xanthomonas*-specific outer membrane protein XAC1347 (OMP$_{Xan}$) that exerts a role in the expression of the type III secretion system for pathogenicity. In this study, we reported that OMP$_{Xan}$ was required for salt stress tolerance and cell membrane integrity, as well as the expression of the *gum* genes for the production of extracellular polysaccharides. Pull-down and yeast two-hybrid assays revealed that OMP$_{Xan}$ interacts with TolB, a substrate of the Tol/Pal membrane protein system. The deletion of *tolB* resulted in similar phenotypic alterations as the OMP$_{Xan}$ mutant in salt stress tolerance, cell membrane integrity, and the expression of *hrpG*, *hrpX*, *hrpD6*, and *hrcC* for pathogenicity. In contrast, the absence of TolB resulted in an increased level of expression of the *gum* genes and the production of extracellular polysaccharides. These results indicate that the interaction of OMP$_{Xan}$ and TolB coordinates multi-faceted mechanisms to manage environmental stress and pathogenicity.

**IMPORTANCE**  The gram-negative *Xanthomoas citri* subsp. *citri* (*Xcc*) is a causal agent of citrus canker, a serious bacterial disease on citrus plants. Our previous research reported that a *Xanthomonas*-specific outer membrane protein XAC1347 (OMP$_{Xan}$) is necessary for type III gene expression. This manuscript provided evidence to show that OMP$_{Xan}$ interacts with Tol/Pal system substrate TolB. Moreover, OMP$_{Xan}$ and TolB are both required for cell membrane integrity, stress adaption, and virulence. The overall results support that OMP$_{Xan}$ and TolB coordinate multi-faceted mechanisms to manage environmental stress and pathogenicity.

**KEYWORDS**  citrus canker, *Xanthomonas*, TolB, outer membrane protein, Tol/Pal, membrane integrity, pathogenicity

**Peer Reviewer** Alexander N. Ignatov, Rossijskij universitet druzby narodov Agrarno-tehnologiceskij insitut, Moscow, Russia

Address correspondence to Huasong Zou, zouhuasong@zjhzu.edu.cn.

The authors declare no conflict of interest.

See the funding table on p. 13.

$X$anthomonas citri subsp. *citri* (*Xcc*) is a causal agent of citrus canker that causes substantial commercial losses of citrus in major production areas (1). The extracellular polysaccharides (EPS) biosynthesis is essential for full virulence in *Xcc* (2). The EPS consist of repeating pentasaccharide units of mannose-(1, 4)-glucuronic acid-(1, 2)-mannose-(1, 3)-cellobiose synthesized by a gene cluster that is composed of the 12 gene members *gumB-M* (3). A mutant that is defective in *gumB* cannot synthesize mature EPS from the pentasaccharide intermediates, which leads to a reduction in the epiphytic survival of this pathogen and a concomitant reduction in disease development (4). A periplasmic alkanesulfonate-binding protein NrtT is a component of the ABC transporter involved in the uptake of sulfur compounds. Deletion of the *nrtT* gene reduces gum production and pathogenicity in the citrus host (5).

A type III secretion system (T3SS) is essential for plant pathogenic bacteria to deliver type III effectors into plant cells (6). The *hrp* gene cluster that encodes T3SS is

indispensable for the hypersensitive response (HR) on nonhosts and pathogenicity on susceptible hosts (6). The *hrp* gene cluster of *Xcc* was first characterized from strain 99–1330, and it is composed of the six operons *hrpA-F* that are under the control of an OmpR family regulator *hrpG* and an AraC-type transcriptional activator *hrpX* (7). The expression of *hrpG*, *hrpX*, and other *hrp* genes is induced *in planta* and the *hrp*-inducing medium XVM2, which mimics the environment of plant intercellular spaces (8, 9). When grown in rich media, the T3SS regulator HrpG is targeted by the Lon protease for proteolysis, which results in the downregulated expression of HrpX and the *hrp* genes. The Lon protease is phosphorylated at serine 654 when *Xcc* interacts with the citrus plant, which leads to attenuation in its proteolytic activity, therefore protecting HrpG from degradation (10). Furthermore, the expression of *Xcc hrp* genes depends on several other factors, including the transcriptional regulators GntR and RsmA/CsrA (11, 12).

The cell envelope of gram-negative bacteria consists of an inner symmetric bilayer of glycerolphospholipids, a peptidoglycan layer located in the periplasmic space, and an outer membrane that is primarily composed of lipopolysaccharides and proteins. A Tol–Pal complex bridges the three layers of the cell envelope, which is composed of five protein components, namely TolQ, TolR, TolA, TolB, and peptidoglycan-associated lipoprotein (Pal) (13). The periplasmic protein TolB interacts with the outer membrane peptidoglycan-associated proteins, Pal, Lpp, and porins (14). Bacteria that are defective in the Tol/Pal complex are more sensitive to antibiotics and detergents, which cause various physiological changes, such as the release of periplasmic proteins from the cells, loss of motility, and abnormal cell division (15–17). In *Escherichia coli*, the Tol/Pal complex is involved in the invasion of bacteriophage particles and the transport of colicin into the cell (18). The mutation of every component of the Tol/Pal complex causes reduced growth, motility, and virulence in *Dickeya dadanti* (*Erwinia chrysanthemi*), the causal agent of soft rot disease in many types of plants (19). The function of the Tol/Pal complex in plant pathogenic *Xanthomonas* strains remains unclear.

Our previous study showed that disruption of the *Xanthomonas*-specific outer membrane protein gene $OMP_{Xan}$ resulted in the loss of pathogenicity in *Xcc*. In addition, the $OMP_{Xan}$ mutant shows a reduced tolerance to copper stress (20). This study reported the roles of $OMP_{Xan}$ in membrane maintenance, EPS production, and stress tolerance. In an effort to elucidate the $OMP_{Xan}$-mediated mechanisms, the Tol/Pal substrate TolB was characterized as a protein that interacts with $OMP_{Xan}$. A mutagenesis analysis of *tolB* showed that *Xcc* requires both the Tol/Pal complex and $OMP_{Xan}$ to manage environmental stress as well as to express the *hrp* genes for pathogenicity. This highlights the functional significance of the interactions between $OMP_{Xan}$ and TolB in *Xanthomonas*.

## RESULTS

### $OMP_{Xan}$ is necessary for stress tolerance and membrane integrity

To detect the role of $OMP_{Xan}$ in stress tolerance, wild-type (WT) 29–1, a deletion mutant ($\Delta OMP_{Xan}$), and complemented strain $C\Delta OMP_{Xan}$ were cultured in nutrient-rich broth (NB) liquid media supplemented with diverse concentrations of NaCl, sorbitol, and sodium dodecyl sulfate (SDS). In comparison to the WT and $C\Delta OMP_{Xan}$, the growth of $\Delta OMP_{Xan}$ was substantially reduced in the media supplemented with NaCl. A rapid decrease in the growth of $\Delta OMP_{Xan}$ was found in the media that had been supplemented with 1.0% NaCl. In the media supplemented with 2% NaCl, $\Delta OMP_{Xan}$ could not replicate to a higher cell density (Fig. 1A). The growth of $\Delta OMP_{Xan}$ rapidly decreased in the media supplemented with 10% sorbitol, and $\Delta OMP_{Xan}$ could not replicate to a higher cell density in 20% sorbitol (Fig. 1B). The growth of $\Delta OMP_{Xan}$ was not significantly affected by supplementation with 0.002% and 0.004% SDS, but it rapidly decreased in the media supplemented with 0.008% SDS (Fig. 1C). Transmission electron microscopy (TEM) was conducted to examine the cell morphology. The WT and complemented strain $C\Delta OMP_{Xan}$ had a normal cell structure with an intact cell membrane. Although the $\Delta OMP_{Xan}$ cell had a clear rod shape, the membrane appeared to be damaged. There were

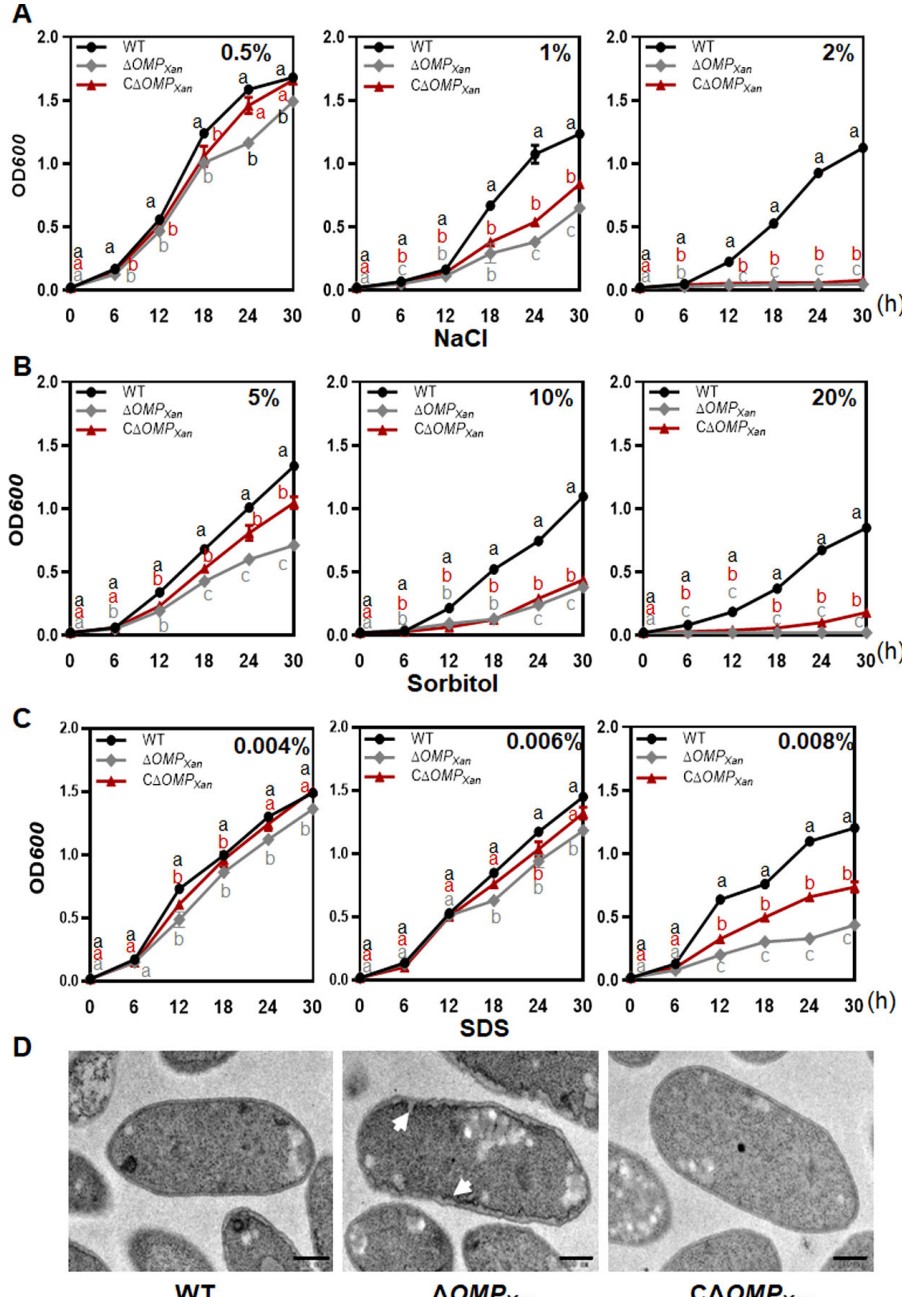

**FIG 1** The stress tolerance and membrane integrity of ΔOMP$_{Xan}$. (A) Involvement of OMP$_{Xan}$ in the resistance to NaCl. *Xanthomonas citri* subsp. *citri* strains were cultured in NB media to the stationary phase, and the bacterial cells were diluted to an OD$_{600}$ of 0.02 with NB media supplemented with 0.5%, 1%, and 2% NaCl. The cell suspensions were cultured at 28°C, and the cell density was evaluated every 6 h post-inoculation. The experiment was repeated three times with the same results. (B) Involvement of OMP$_{Xan}$ in the resistance to sorbitol. The experiment was performed as described in (A). The cells were suspended using NB media supplemented with 5%, 10%, and 20% sorbitol. The experiment was repeated three times with the same results. (C) Involvement of OMP$_{Xan}$ in the resistance to SDS. The experiment was performed as described in (A). The cells were suspended using NB media supplemented with 0.004%, 0.006%, and 0.008% SDS. The experiment was repeated three times with same results. (D) Transmission electron micrograph of the ΔOMP$_{Xan}$ cells. The cells were cultured in NB media and subjected to this analysis at an exponential growth stage. The changes in the cell membrane are indicated by arrows. Scale bar = 200 μM.

many sawtooth-like structures in the cell membrane, which demonstrated that the cell membrane was destroyed in the absence of OMP$_{Xan}$ (Fig. 1D).

## OMP$_{Xan}$ plays a positive role in the biosynthesis of exopolysaccharide

ΔOMP$_{Xan}$ had a reduced production of EPS in that the colony was drier than the WT when incubated on nutrient-rich agar plates. This suggested that the deletion of OMP$_{Xan}$ affected the biosynthesis of EPS. To further verify this conclusion, the EPS was quantified from the Δ*OMP$_{Xan}$* cells cultured in NB and nutrient yeast glycerol broth (NYGB) media. In comparison to WT 29–1, the production of EPS in the mutant ΔOMP$_{Xan}$ decreased by 29% ($P < 0.05$) and 33% ($P < 0.05$) in NB and NYBG, respectively (Fig. 2A and B). To determine if the level of expression of *gum* changed, a *gumB* promoter-β-glucuronidase (GUS) fusion was constructed and expressed in Δ*OMP$_{Xan}$* to monitor the expression of *gum* operon (Fig. S1). The GUS activity was reduced remarkably in Δ*OMP$_{Xan}$*, which indicated that the *gum* operon was downregulated (Fig. 2C). The *gumD* and *gumE* genes in the operon were selectively used for a quantitative real-time reverse transcription PCR (qRT-PCR) analysis. The results showed that transcript levels of *gumD* and *gumE* were reduced by 57% ($P < 0.01$) and 54% ($P < 0.01$) compared with those in the WT, respectively (Fig. 2D). This demonstrated that the deletion of *OMP$_{Xan}$* resulted in the downregulation of *gum* gene expression to produce the EPS.

## OMP$_{Xan}$ interacts with the Tol/Pal system substrate TolB

The prokaryotically expressed GST-OMP$_{Xan}$ was used for a glutathione S-transferase (GST) pull-down analysis followed by overnight co-incubation with the total proteins of *Xcc*. The proteins pulled down by GST-OMP$_{Xan}$ were scored by SDS-PAGE electrophoresis in comparison to the proteins pulled down by the GST tag (Fig. S2). The proteins that interacted with OMP$_{Xan}$ were characterized by mass spectrometric analyses. Three proteins with high coverage and matches were identified, including the membrane periplasmic TolB (XAC3142), elongation factor Tu (XAC0957 and XAC0970), and phosphoglucosamine mutase (XAC2714). Among them, TolB showed the highest coverage (68%) and number of matches (111/79) (Table S1). The interaction between OMP$_{Xan}$ and TolB was subsequently verified by DUAL membrane yeast two-hybridization. The yeast strains that harbored pPR3-N/pTSU2-APP or pPR3-N/pBT-OMP$_{Xan}$ could not grow on SD/-Met-Trp-Leu-Ade, whereas the yeast that harbored pPR3-TolB/pBT-OMP$_{Xan}$ grew well. Furthermore, positive LacZ activity was found in the yeast that harbored pPR3-TolB/pBT-OMP$_{Xan}$ (Fig. 3A). The interaction was further verified by maltose-binding protein (MBP) pull-down assays (Fig. 3B).

## Δ*tolB* shows reduced virulence with the downregulation of *hrp* genes

To study the role of *tolB* in *Xcc*, two steps of recombination were used to conduct deletion mutagenesis of the *tolB* coding region. The primer set tolB1.F/tolB2.R was used for this experiment (Table S2), and the mutant was identified by PCR amplification of a 756 bp fragment produced from the deletion mutant, whereas a 1,899 bp fragment was generated from the WT (Fig. S3). The 1,500 bp DNA fragment that was composed of the *tolB* coding region and its promoter was cloned into pBBR1MCS-5, which resulted in a recombinant pBB-*tolB* for the complementation analysis. The deletion of *tolB* led to reduced replication in the NB media (Fig. 4A).

The WT 29–1, Δ*tolB*, and the complemented strain CΔ*tolB* were inoculated on citrus by infiltration to examine any deviation in virulence. In comparison with the WT, Δ*tolB* caused no symptoms on the leaves at 5 d post-inoculation (Fig. 4B). The Δ*tolB* cells collected from the citrus leaves were reduced by 20-fold compared with those from the WT (Fig. 4C). In addition, the mutant had lost the ability to induce an HR on tomato (*Lycopersicon esculentum*) plants (Fig. 4D). The *hrpG* and *hrpX* promoter GUS fusions were expressed in Δ*tolB* to quantify the GUS activity in the XVM2 *hrp*-inducing medium. The results showed that the GUS activity that was driven by the *hrpG* promoter was reduced

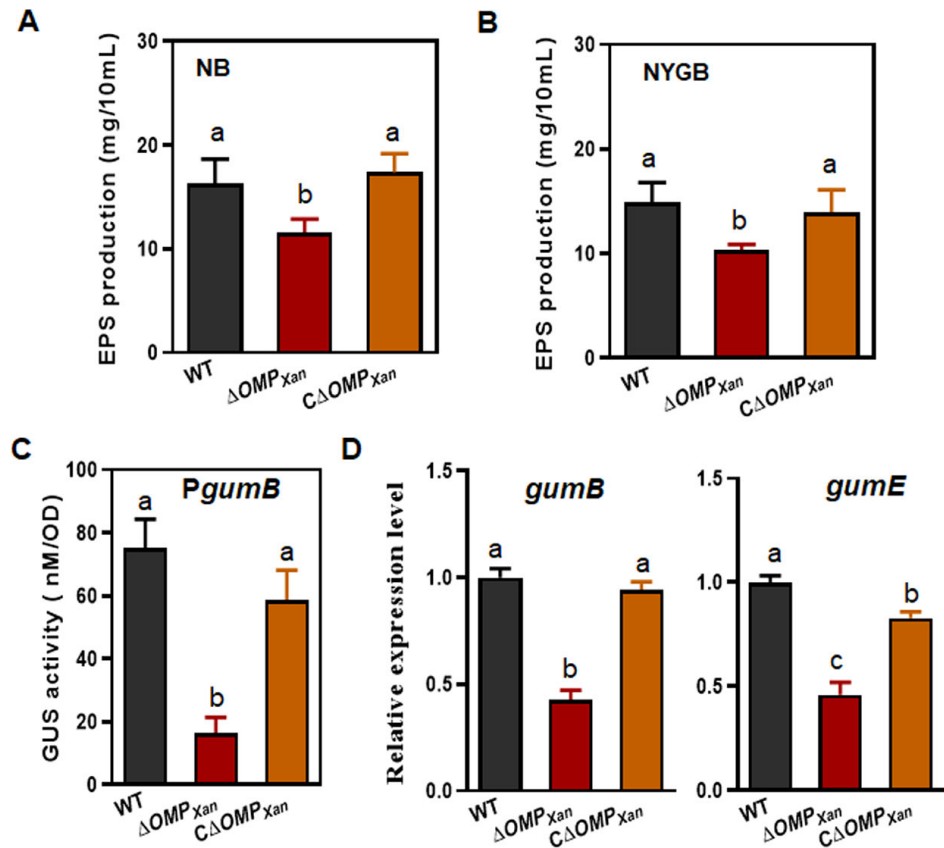

**FIG 2** The reduced production of EPS by $\Delta OMP_{Xan}$. (A) Quantification of the production of EPS by $\Delta OMP_{Xan}$ cultured in NB media. The EPS was precipitated from a 10 mL cell suspension of an $OD_{600}$ of 1.0 by adding 30 mL of ethanol. (B) Quantification of the production of EPS by $\Delta OMP_{Xan}$ cultured in NYGB media. The EPS was precipitated from a 10 mL cell suspension of $OD_{600}$ of 1.0 by adding 30 mL of ethanol. (C) GUS activity triggered by the *gumB* promoter in $\Delta OMP_{Xan}$. The GUS activity was measured from the cells of a 2 mL bacterial suspension ($OD_{600} = 1.0$) using *p*-nitrophenol-β-D-glucuronide as the substrate. The activity was counted as nmol product $min^{-1}$ $OD_{600}$ $unit^{-1}$. (D) qRT-PCR analysis of the transcript levels of *gumB* and *gumE*. RNA was extracted from *Xcc* cells cultured in NB media. The level of expression in the WT was established as 1, and the levels of expression in the mutant and complemented strains were calculated by comparison. All the experiments were repeated three times. The different letters above the columns indicate a significant difference at *P* = 0.05 (one-way ANOVA). ANOVA, analysis of variance; EPS, extracellular polysaccharides; GUS, β-glucuronidase; qRT-PCR, quantitative real-time reverse transcription PCR; WT, wild type.

by a remarkable amount; moreover, the GUS activity driven by the *hrpX* promoter was reduced by 41% (*P* < 0.05) compared with the WT (Fig. 4E and F). This suggested that *tolB* is required for the expression of the two *hrp* gene regulators. To verify this further, qRT-PCR was conducted to assess the transcription of *hrpG* and *hrpX*, as well as *hrpD6* and *hrcC*, *in planta*. The transcript levels of the four genes were downregulated by 35%, 80%, 67%, and 46% in $\Delta tolB$, respectively (*P* < 0.01) (Fig. 4G through J).

## $\Delta tolB$ shows reduced tolerance to stress conditions

$\Delta tolB$ was more sensitive to stress stimuli when NaCl, sorbitol, and SDS were added to the NB media, which is similar to the $OMP_{Xan}$ mutant. The growth of $\Delta tolB$ was rapidly reduced in the media supplemented with 1% NaCl, 10% sorbitol, and 0.008% SDS. Moreover, $\Delta tolB$ did not grow to a higher cell density in 2% NaCl or 20% sorbitol (Fig. 5A through C). The necessity of TolB for cell membrane integrity was studied accordingly by scanning electron microscopy. In comparison with the WT, the membrane integrity of $\Delta tolB$ was severely affected in a manner similar to that of the $OMP_{Xan}$ mutant (Fig. 5D).

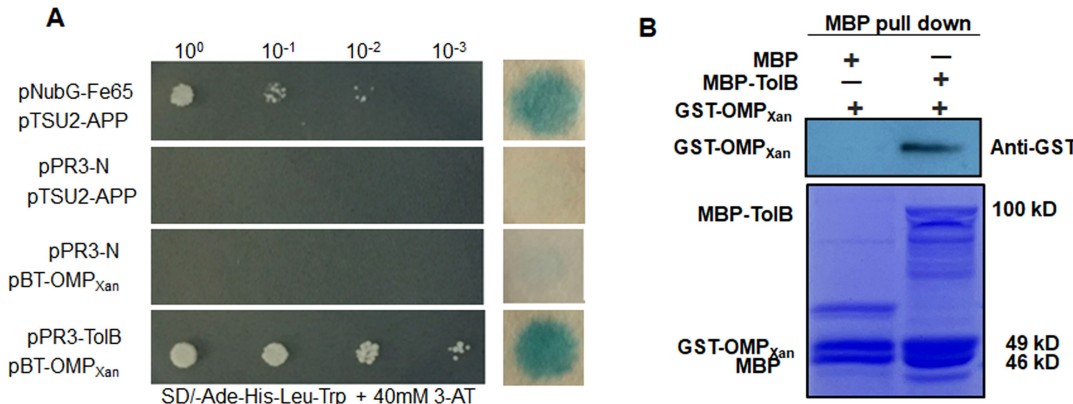

**FIG 3** OMP$_{Xan}$ interacts with TolB. (A) Y2H shows the interaction between OMP$_{Xan}$ and TolB. pPR3-TolB and pBT-OMP$_{Xan}$ were co-transformed into yeast cells and screened on synthetic dextrose media that lacked leucine and tryptophan (SD/-Leu-Trp) and was supplemented with 40 mM 3-amino-1,2,4-triazole (3-AT). A single yeast colony was then selected for serial dilution and grown on SD/-Ade-His-Leu-Trp to examine the interaction. The yeast co-transformed with pNubG-Fe65 and pTSU2-APP served as a positive control. The yeast co-transformed with pPR3-N and pTSU2-APP served as a negative control. The β-galactosidase activity of the transformants was assayed on a filter soaked with Z buffer that contained 20 µg mL$^{-1}$ X-gal after the yeast cells were subjected to five freeze-thaw cycles using liquid nitrogen. (B) MBP pull-down assay of the interaction between OMP$_{Xan}$ and TolB. The recombinant GST-OMP$_{Xan}$ and MBP-TolB proteins were used for the MBP pull-down assays. The MBP protein served as the negative control. The pull-down of GST-OMP$_{Xan}$ was verified by anti-GST immunoblotting. The experiment was repeated three times with similar results. GST, glutathione-S-transferase; MBP, maltose-binding protein; Y2H, yeast-two hybrid.

## The production of EPS is increased in Δ*tolB*

The production of EPS of Δ*tolB* increased by 1.7-fold ($P < 0.01$) and 1.6-fold ($P < 0.05$) compared with that of the WT in NB and NYGB, respectively (Fig. 6A and B). This suggests that *tolB* suppressed the production of EPS. The expression of *gum* was studied in the cells cultured in NB media. The *gumB* promoter-GUS fusion was expressed in Δ*tolB* to examine the expression of *gum* operon. The results showed that the GUS activity triggered by the *gumB* promoter increased by 29.5% compared with that in the WT ($P < 0.01$) (Fig. 6C). Simultaneously, the transcript levels of *gumB* and *gumE* increased by 1.8-fold ($P < 0.01$) and 1.4-fold ($P < 0.01$), respectively (Fig. 6D and E). Therefore, the loss of *tolB* led to the increased production of EPS and the enhanced expression of *gum* genes.

## DISCUSSION

Our previous study reported the necessity of OMP$_{Xan}$ for copper homeostasis and the expression of *hrp* genes (20). This study extensively studied its role in cell membrane integrity, salt stress tolerance, and EPS production. Most importantly, OMP$_{Xan}$ was found to interact with the Tol/Pal complex subunit TolB. A mutagenesis analysis revealed that *tolB* was also required for cell membrane morphology, stress resistance, and virulence. These data support the concept that OMP$_{Xan}$ acts as a bridge to maintain the stability of TolB between the periplasm and outer membrane, which helps improve the integration of Tol/Pal complex into the cell membrane in *Xcc*. This conclusion expanded our knowledge of the assembly and function of the Tol/Pal complex.

Xcc possesses the known T3SS that is responsible for releasing a repertoire of effectors into its host cell (21). Based on genetic structure and regulatory pattern, the *Xcc hrp* gene cluster that encodes T3SS is extremely homologous to ones from the other pathovars of *Xanthomonas* (22). Our previous research revealed that *hrcV*, *hrpD6*, *hrpF*, and *hrpB1* are downregulated in mutant Δ*OMP$_{Xan}$* (20). We used P*hrpG*-and P*hrpX*-GUS fusions to show that the *hrp* gene regulators *hrpG* and *hrpX* were downregulated in Δ*OMP$_{Xan}$* (Fig. S4). This demonstrated that the loss of OMP$_{Xan}$ led to the downregulation of the entire *hrp* gene cluster. The most probable reason is that OMP$_{Xan}$ is involved in perceiving specific signals to activate the regulatory cascade for the *hrp* genes. This is

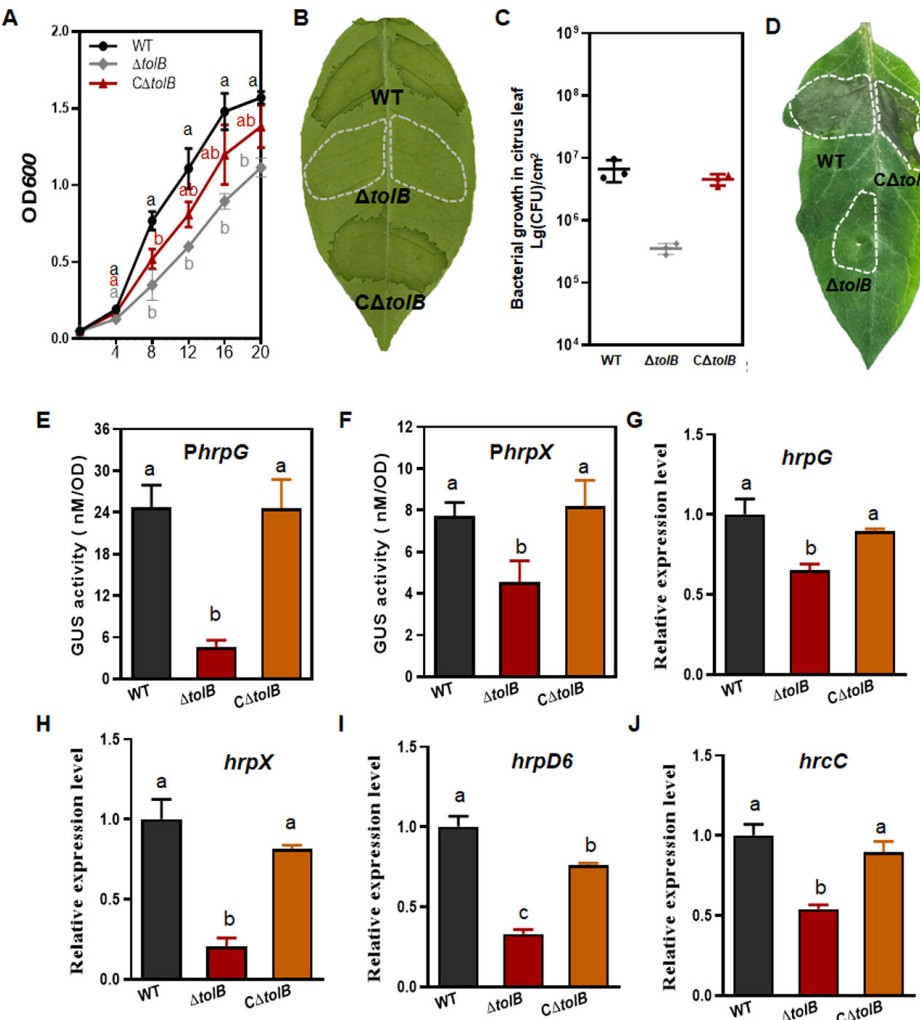

**FIG 4** The phenotypes of the nonpolar deletion mutant Δ*tolB* in grapefruit and tomato. (A) Growth of Δ*tolB* in NB media. The cultured cells were prepared to an initial concentration of $OD_{600} = 0.05$ in NB media. The growth was assayed every 4 h post-cultivation at 28°C. (B) Citrus canker caused by Δ*tolB* on grapefruit. Inoculum of $OD_{600} = 0.3$ was infiltrated into the plant leaves, and the phenotype was recorded at 5 d post-inoculation. (C) Growth of Δ*tolB* in the citrus leaves. The cells of $10^8$ cfu mL$^{-1}$ were infiltrated into the citrus leaves. At 2 days post-inoculation, the bacteria were recovered from the leaves and counted on NA plates. Error bars represent the SD from three independent experiments. (D) The hypersensitive response on tomato. Cells of $10^8$ cfu mL$^{-1}$ were infiltrated into the tomato leaves. The hypersensitive response was scored at 24 h post-inoculation. (E, F) *hrpG* and *hrpX* promoter activities in Δ*tolB*. The GUS activities were quantified in Δ*tolB* that harbored P*hrpG*-GUS or P*hrpX*-GUS fusions cultured in the *hrp*-inducing medium XVM2. The GUS activity was measured with *p*-nitrophenol-β-D-glucuronide as substrate and counted as nmol product per min per OD unit. Changes in the pattern of expression were revealed by comparison with the WT and complemented strains. (G–H) qRT-PCR analysis of the transcript levels of *hrpG*, *hrpX*, *hrpD6, and hrcC*. RNA was extracted from the *Xcc* cells inoculated in citrus plants at 3 days post-inoculation. The level of expression in the WT was established as 1, and the levels of expression in the mutant and complemented strains were calculated by comparison with that of the WT. All the experiments were repeated three times. The different letters above the columns indicate a significant difference at $P = 0.05$ (one-way ANOVA). ANOVA, analysis of variance; GUS, β-glucuronidase; NA, nutrient-rich agar; qRT-PCR, real-time quantitative reverse transcription PCR; WT, wild type.

different from the periplasmic protein VrpA that physically interacts with the periplasmic T3SS components HrcJ and HrcC. The absence of VrpA substantially affects the efficiency of the secretion of effectors through T3SS (23).

The Tol/Pal system is important for bacterial motility, cell survival, adhesion to the host surface, and bacterial virulence. In enterohemorrhagic *E. coli*, the efficiency of the

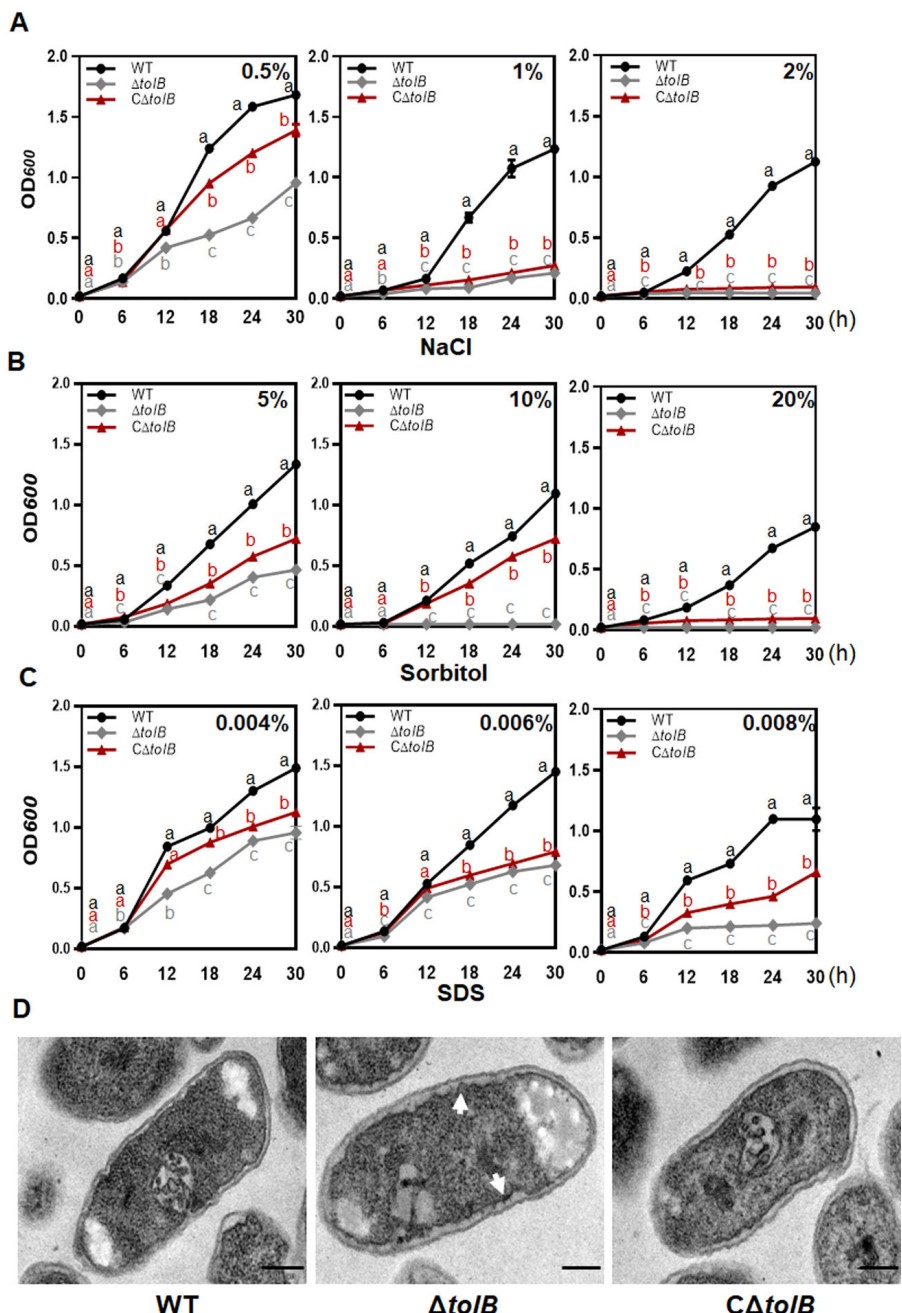

**FIG 5** The stress tolerance and membrane integrity of Δ*tolB*. (A) Involvement of *tolB* in resistance to NaCl. *Xanthomonas citri* subsp. *citri* strains were cultured in NB media to their stationary phase, and the bacterial cells were diluted to $OD_{600} = 0.02$ with NB media supplemented with 0.5%, 1%, and 2% NaCl. The cell suspensions were cultured at 28°C, and the cell density was evaluated every 6 h post-inoculation. The experiments were repeated three times with the same results. (B) Involvement of *tolB* in resistance to sorbitol. The experiment was performed in the same manner as in (A). The cells were suspended using NB media supplemented with 5%, 10%, and 20% sorbitol. The experiment was repeated three times with the same results. (C) Involvement of *tolB* in resistance to SDS. The experiment was performed in the same manner as in (A). The cells were suspended using NB media supplemented with 0.004%, 0.006, and 0.008 SDS. The experiment was repeated three times with the same results. (D) Transmission electron micrograph of Δ*tolB* cells. The cells were cultured in NB media and subjected to this analysis at the exponential growth stage. The changes in the cell membrane are indicated by arrows. Scale bar = 200 µM.

secretion of the T3SS proteins EspA/B is significantly reduced in *tolB* mutants; moreover, the transcription of *EspA/B* showed no difference from that of the WT (24). The emerging

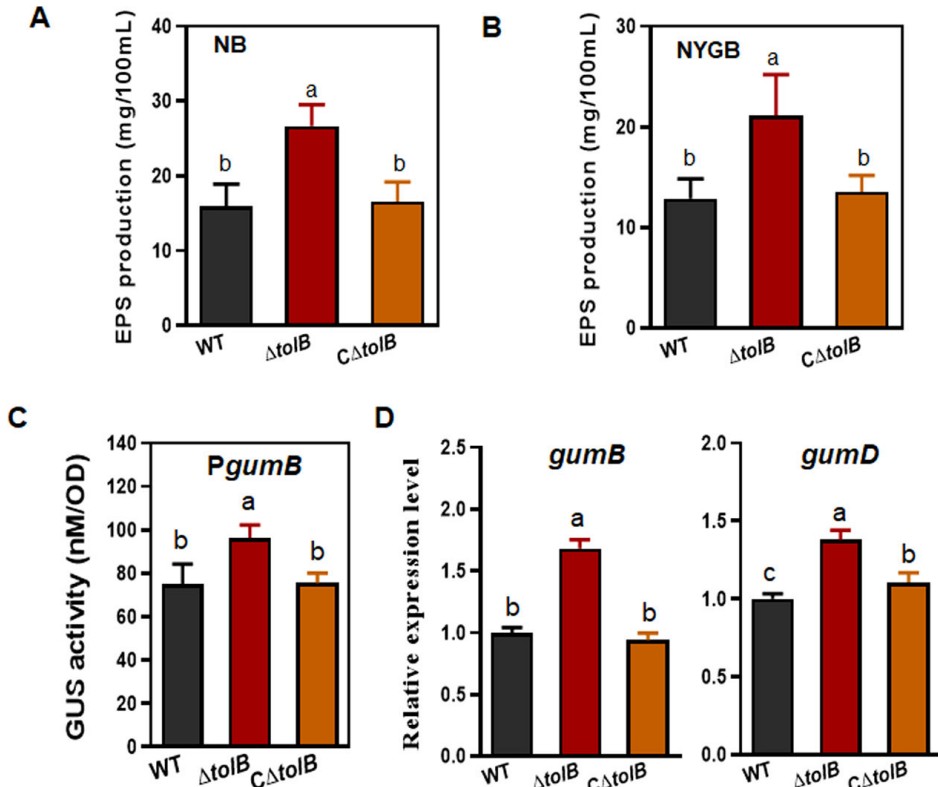

**FIG 6** The increased production of EPS by Δ*tolB*. (A) Quantification of the production of EPS by Δ*tolB* cultured in NB media. The EPS was quantified from a 10 mL cell suspension of $OD_{600}$ = 1.0 by adding 30 mL of alcohol. (B) Quantification of the production of EPS by Δ*tolB* cultured in NYGB media. The EPS was precipitated as described in (A). (C) GUS activity was triggered by the *gumB* promoter in Δ*tolB*. The GUS activity was measured from cells of 2 mL of a bacterial suspension ($OD_{600}$ = 1.0) using *p*-nitrophenol-β-D-glucuronide as the substrate. The activity was counted as nmol product $min^{-1}$ $OD_{600}^{-1}$. (D) qRT-PCR analysis of the transcript levels of *gumB* and *gumD*. The RNA was extracted from the *Xcc* cells cultured in NB media. The level of expression in the WT was established as 1, and the levels of expression in the mutant and complemented strains were calculated by comparison with the WT. All the experiments were repeated three times. The different letters above the columns indicate a significant difference at *P* = 0.05 (one-way ANOVA). ANOVA, analysis of variance; EPS, extracellular polysaccharides; GUS, β-glucuronidase; NB, nutrient broth; NYGB, nutrient yeast glycerol broth; qRT-PCR, real-time quantitative reverse transcription PCR; *Xcc*, *Xanthomonas citri* subsp. *citri*; WT, wild type.

pathogen *Xylella fastidiosa* that is limited to the plant xylem dynamically expresses TolB and Pal in biofilm cells, which are expressed at higher levels at 3, 5, and 20 days during biofilm development, which corresponds to the initial bacterial cell surface adhesion, irreversible adhesion to the surface, and mature biofilm, respectively (25). In *D. dadanti* (*E. chrysanthemi*), the mutation of every component of the five Tol/Pal proteins distinctively reduces the virulence of this pathogen (19). In this study, the mutagenesis of *Xcc tolB* impaired the ability of bacteria to cause disease on citrus plants and induce HR on nonhost tomato plants, which exhibits a similar phenotypic alteration with the T3SS mutants. Most importantly, *hrpG* and *hrpX*, as well as *hrpD6* and *hrcC*, were all downregulated in Δ*tolB*. This apparently demonstrated that *tolB* is required for the expression of T3SS, which is similar to OMP$_{Xan}$.

The Tol–Pal complex is assembled by precise protein-protein interactions among the five substrates. TolA, TolQ, and TolR are inner membrane proteins; TolB is periplasmic; and Pal is an abundant outer membrane lipoprotein (26). An interaction between the C-terminal domain of TolA and the N-terminal domain of TolB is essential to building a steady system (14). Furthermore, the periplasmic TolB interacts with Pal, which is located on the outer membrane (27). Our research in this study verified that *Xcc* TolB interacts

with the outer membrane protein $OMP_{Xan}$. Although the $OMP_{Xan}$-interacting domain has not been clearly elucidated in TolB, the interaction between TolB and $OMP_{Xan}$ appeared to be required for *Xcc* to maintain membrane integrity. The deletion of either $OMP_{Xan}$ or *tolB* substantially reduced the bacterial tolerance to stress conditions and, in particular, destroyed the cell membrane morphology as shown by TEM.

As in other pathovars of *Xanthomonas*, *Xcc* synthesizes EPS through a conserved *gumB-M* cluster of 12 genes that are responsible for the synthesis and polymerization of the lipid intermediate. Those *gum* genes are primarily expressed as an operon that relies on a promoter upstream of the first gene *gumB*, which encodes the outer membrane export protein for EPS (28, 29). The disruption of *Xcc* outer membrane protein B (OprB), a family of glucose transporters, leads to the increased production of EPS, as well as its carbohydrate intermediates (30). Some other outer membrane proteins have been extensively reported for their involvement in EPS export, such as TolC in *Sinorhizobium meliloti* (31) and PelC in *Pseudomonas aeruginosa* (32). Among all the phenotypic alterations examined in this study, the production of EPS was the sole one that differed between $\Delta OMP_{Xan}$ and $\Delta tolB$. It appeared that this was caused by a differential pattern of expression of the *gum* genes. Based on a *gumB* promoter-GUS fusion and qRT-PCR analysis, it was verified that the *gum* genes were upregulated in the mutant $\Delta tolB$ and downregulated in $\Delta OMP_{Xan}$. Thus, $OMP_{Xan}$ relied on other mechanisms to affect the production of EPS that were independent of the Tol/Pal complex.

The outer membrane protein $OMP_{Xan}$ is only found in *Xanthomonas* strains and is therefore a genus-specific trait. Since it shares extremely high homology, it is assumed that $OMP_{Xan}$ has a similar function in diverse *Xanthomonas* strains. An insertion mutagenesis strategy was used to create an $OMP_{Xan}$ mutant in the plant pathogenic bacterium *X. campestris* pv. *campestris*, the causal agent of black rot on crucifers. The *X. campestris* pv. *campestris* mutant was deficient in EPS production. Moreover, it was unable to cause black rot disease in wild cabbage (*Brassica oleracea*) (Fig. S5). Although a limited phenotypic examination was performed on the mutant, the EPS production and virulence assay strongly indicated that $OMP_{Xan}$ of *X. campestris* pv. *campestris* had the same function as that in *Xcc*. The roles of $OMP_{Xan}$ in other plant pathogenic pathovars of *Xanthomonas* merit further elucidation.

In conclusion, this study reported that $OMP_{Xan}$ interacts with the periplasmic substrate TolB of the Tol/Pal system. Both $OMP_{Xan}$ and TolB are required for stress tolerance, membrane integrity, and the expression of *hrp* genes. These results highlight the role of $OMP_{Xan}$ and the Tol/Pal system in *Xanthomonas* strains.

## MATERIALS AND METHODS

### Bacterial strains, plasmids, and growth conditions

The bacterial strains and plasmids used in this study are listed in Table S2. The *Xcc* strains were cultivated in NB media or NB added with 1.5% agar (NA) at 28℃. Yeast strain AH109 was grown in yeast extract peptone dextrose (YPD) media at 30℃ (33). The *E. coli* strains were cultured in LB media at 37℃. Antibiotics were applied at the following concentrations: ampicillin (Amp), 100 µg mL$^{-1}$, kanamycin (Km) at 50 µg mL$^{-1}$, spectinomycin (Sp) at 50 µg mL$^{-1}$ and gentamycin (Gm) at 10 µg mL$^{-1}$.

### Analysis of bacterial growth under stress conditions

The *Xcc* strains were cultured in NB broth at 28℃ for 36 h and sub-cultured (1:100) in 5 mL fresh NB broth until the $OD_{600}$ reached 1.0. The cell culture was sub-cultured (1:50) in 10 mL of NB supplemented with gradient concentrations of NaCl (0.5%, 1%, and 2%), sorbitol (5%, 10%, and 20%), and SDS (0.004%, 0.006%, and 0.008%). The growth rates were assessed by measuring the $OD_{600}$ values every 6 h after sub-culturing. All the experiments were repeated three times.

## Transmission electron microscopy

The *Xcc* cells cultured in NB media were fixed with 3% glutaraldehyde in 0.1 M potassium phosphate buffer (pH 7.2) at room temperature for 4 h. After the cells had been washed twice with the same buffer, they were post-fixed in 2% osmium tetroxide in this buffer at room temperature for 4 h (34). The samples were then dehydrated in an ethanol series and embedded in Spurr's resin. Thin sections (80–100 nm thick) were cut with a diamond knife, collected on 200 mesh copper grids, stained with 2% uranyl acetate, and post-stained with lead citrate. The cells were then examined under a TEM (SU8020; Hitachi, Ltd., Tokyo, Japan) in scanning TEM mode. The experiments were repeated three times.

## Quantification of the production of EPS

The EPS production was quantified by the ethanol precipitation method (30). The *Xcc* strains were incubated in 10 mL NB or NYGB for 72 h. After the cells were removed by centrifugation (5,000 g for 10 min), three volumes of ethanol were added to the supernatants. The precipitated EPS was pelleted by centrifugation (10,000 g for 10 min), dried, and weighed. Three independent replicates were used for each strain. The test was performed three times independently.

## GST pull-down and mass spectrometry

$OMP_{Xan}$ was cloned into pET41a(+) at the *EcoR*I and *Sal*I sites to fuse with the GST. The resulting construct was transformed into BL21 (DE3) cells for induction with 1.0 mM isopropyl-β-D-thiogalactopyranoside (IPTG). GST-$OMP_{Xan}$ was purified by a glutathione resin. The total protein of *Xcc* 29–1 was extracted using a Bacterial Protein Extraction Kit (Sangon Biotech, Shanghai, China). A total of 3 μg GST-$OMP_{Xan}$ and 5 μg of *Xcc* total proteins were co-incubated overnight at 4°C. The mixture was then purified using the glutathione resin. The eluted protein samples were analyzed by 12% SDS-PAGE. A comparison with the GST control was used to excise specific bands from GST-$OMP_{Xan}$ from the gel and then subject them to a mass spectrometry analysis by Huada Gene Technology Co., Ltd. (Shenzhen, China). The proteins with >50% average coverage from the three replicates were considered to be putative interacting candidates. GST, glutathione-S-transferase.

## Yeast two-hybrid assay

The *tolB* gene was cloned into the pPR3-N vector, and $OMP_{Xan}$ was cloned into the pBT3-SET vector. The two resulting constructs were then co-transformed into the yeast strain NMY51. The positive transformants were screened on SD/-Ade/-Trp/-His and SD/-Ade/-Leu/-Trp/-His/ supplemented with 40 mM 3-amino-1,2,4-triazole (3-AT). Subsequently, the interaction between $OMP_{Xan}$and TolB was confirmed by incubation on SD/-Ade/-Leu/-Trp/-His/ plates supplemented with 40 mM 3-AT. The cultured transformants were prepared to a cell density of $OD_{600}$ = 1.0 and then diluted to a 10-fold series. For each concentration series, 2 μL of suspensions were spotted on plates and incubated for 4 days. pNubG-Fe65/pTSU2 APP and pPR3-N/ pTSU2-APP were used as the positive and negative control, respectively (35). The β-galactosidase assay was conducted on a filter soaked with Z buffer that contained 20 μg mL$^{-1}$ X-gal after the yeast cells had been subjected to five freeze-thaw cycles using liquid nitrogen.

## MBP pull-down assay

*tolB* was cloned into pMAl-C4X at the *EcoR*I and *Sal*I sites to fuse with the MBP. The recombinant construct was transformed into BL21 (DE3) cells for induction with 1.0 mM IPTG. MBP-TolB was purified by amylose affinity chromatography (Genescript, Nanjing, China). A total of 3 μg of GST-$OMP_{Xan}$ and MBP-TolB were incubated overnight at 4°C. Proteins eluted from amylose resins were boiled in 1× SDS loading buffer and subjected

to electrophoresis on a 12% SDS-PAGE gel. The immunoblotting was performed with an anti-GST antibody. The experiments were repeated three times.

## Construction of the Δ*tolB* mutant and complemented strains

A 217 bp fragment of the 5′ terminus of *tolB* was amplified by PCR using the primer set to lB1.F/tolB1.R, and a 539 bp DNA fragment composed of the *pal* gene was amplified by the primer set tolB2.F/tolB2.R (Table S3). The two fragments were inserted into the suicide vector pKMS1 at the *Bam*HI and *Sal*I sites, which resulted in pKMS-tolB (Table S2). The recombinant plasmid was introduced into the WT *Xcc* 29–1 to generate deletion mutants by two steps of homologous recombination (33). After selection on NA plates supplemented with 10% sucrose, the colonies were subjected to PCR analysis with the primer set tolB1.F/tolB2.R. The Δ*tolB* mutant was identified as a 756 bp PCR product with a deletion of the 1103 bp *tolB* coding sequence. For complementation analysis, the primer pair CtolB.F/CtolB.R was used to amplify a 1,500 bp DNA fragment that contained *tolB* and the promoter region (Table S2). The PCR product was cloned into vector pBBR1MCS-5, which resulted in pBB-tolB. pBB-tolB was transformed into the Δ*tolB* mutant for complementation analysis.

## Pathogenicity, hypersensitive response, and replication assays *in planta*

The cultured *Xcc* cells were suspended in sterile distilled water to a final concentration of $10^8$ cfu $mL^{-1}$ ($OD_{600}$ = 0.3). For the pathogenicity assay, bacterial suspensions were injected into fully expanded grapefruit (*Citrus × paradisi*) leaves with a needleless syringe. The disease symptoms were scored and photographed at 5 days post-inoculation. For the growth assay *in planta*, 0.8 cm diameter leaf discs were collected with a cork borer. The cells were completely ground in 1 mL of sterile $ddH_2O$ to release them from the leaf discs. Serial dilutions of the suspension were spread on NA plates, and the individual colonies were recorded to determine the cfu per $cm^2$ leaf. The SD was calculated based on the colony counts for three triplicate disks that were sampled from each of the three samples per time point per inoculum. The cell suspension ($10^8$ cfu $mL^{-1}$) was infiltrated into tomato leaves to examine the HR reaction. The HR was viewed 24 h post-inoculation. Each test was repeated three times.

## GUS activity assay

The *hrpG*, *hrpX*, and *gumB* promoters were cloned into the pRG960 vector to fuse with GUS (Table S2). P*hrpG*-GUS, P*hrpX*-GUS, and P*gumB*-GUS were introduced into the *Xcc* strains by electrotransformation. The strains that harbored P*hrpG*-GUS or P*hrpX*-GUS were cultured in NB. After the cells had been collected by centrifugation (5,000 g for 10 min), they were re-suspended to $OD_{600}$ = 1.0 with the *hrp*-inducing medium XVM2 (36). The GUS activity was assayed followed by 6 h post-incubation at 28°C. The strains that harbored P*gumB*-GUS were cultured in NB and adjusted to $OD_{600}$ = 1.0. The promoter-driven GUS activity was quantified as previously described (37). The GUS activity was measured with *p*-nitrophenol-β-D-glucuronide as the substrate and counted as nanomoles of product per min per OD unit. The average and standard deviation of three replicate measurements were calculated. The experiments were repeated three times.

## Real-time quantitative reverse transcription PCR (qRT-PCR)

RNA was extracted from the *Xcc* cells using an RNAprep pure Kit for Cell/Bacteria (Tiangen Biotech, Beijing, China). A Plant RNA Kit (Omega Bio-tek, Norcross, GA, USA) was used to isolate RNA from the grapefruit leaves inoculated with *Xcc*. A total of 2 μg of total RNA was reverse transcribed into single-stranded cDNA synthesized with HiScript QRT SuperMix (Vazyme, Nanjing, China). All the primers used for qRT-PCR are listed in Table S3. The PCR thermal cycle conditions were as follows: 95°C for 5 min, followed by 40 cycles of denaturation at 95°C for 5 s, and 60°C for 20 s. The expression of *gyrA*

was used as the internal control. All the experiments included three biological replicates, each with three technical replicates.

## Sequence analysis

The genome sequence information of *Xcc* 306 and 29–1 was obtained from the NCBI (AE008923 and CP004399). The promoter sequence was predicted using http://www.fruitfly.org/seq_tools/promoter.html.

## ACKNOWLEDGMENTS

This work was supported by the National Natural Science Foundation of China (grant number 31872919).

L.W. Data curation, investigation, and writing – original draft | X.Z. Investigation and methodology | H.L. Formal analysis and investigation | X.H. Formal analysis and methodology | S.Z. Supervision and writing – review and editing | Q.X. Conceptualization and writing – review and editing | Z.L. writing – review and editing | H.Z. Formal analysis, conceptualization, validation, writing – original draft, and writing – review and editing.

## AUTHOR AFFILIATION

[1]School of Life Sciences and Health, Huzhou College, Huzhou, Zhejiang, China

## AUTHOR ORCIDs

Huasong Zou  http://orcid.org/0000-0001-5758-1975

## FUNDING

| Funder | Grant(s) | Author(s) |
| --- | --- | --- |
| MOST | National Natural Science Foundation of China (NSFC) | 31872919 | Huasong Zou |

## AUTHOR CONTRIBUTIONS

Leilei Wu, Data curation, Investigation, Writing – original draft | Xueyan Zhu, Investigation, Methodology | Hang Long, Formal analysis, Investigation | Xinyu Huang, Formal analysis, Methodology | Shuying Zhu, Supervision, Writing – review and editing | Qi Xue, Conceptualization, Writing – review and editing | Ziao Li, Writing – review and editing | Huasong Zou, Conceptualization, Formal analysis, Validation, Writing – original draft, Writing – review and editing

## DATA AVAILABILITY

The data that support the findings of this study are available in the supplementary material of this article.

## ADDITIONAL FILES

The following material is available online.

### Supplemental Material

**Supplemental figure captions (Spectrum02521-24-s0001.docx).** Figure captions for Fig. S1 to S5.
**Figure S1 (Spectrum02521-24-s0002.tif).** Schematic diagram showing the cloned *gum* promoter.
**Figure S2 (Spectrum02521-24-s0003.tif).** SDS-PAGE analysis of the final eluted proteins in the GST pull-down.
**Figure S3 (Spectrum02521-24-s0004.tif).** Deletion mutagenesis of tolB.

**Figure S4 (Spectrum02521-24-s0005.tif).** *hrpG* and *hrpX* promoter activities in Δ*OMP~Xan~*.

**Figure S5 (Spectrum02521-24-s0006.tif).** Phenotypic analysis of the *OMP~Xan~* mutant of *Xanthomonas campestris* pv. *campestris*.

**Table S1 (Spectrum02521-24-s0007.docx).** Characterized OMP~Xan~-interacting proteins from *Xanthomonas citri* subsp. *citri*

**Table S2 (Spectrum02521-24-s0008.docx).** Bacterial strains and plasmids used in this study.

**Table S3 (Spectrum02521-24-s0009.docx).** Primers used in this study.

## Open Peer Review

**PEER REVIEW HISTORY (review-history.pdf).** An accounting of the reviewer comments and feedback.

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
