## [Reviewer comments · Microbiology Spectrum]

Microbiology Spectrum

Xanthomonas citri subsp. citri requires a genus-specific outer membrane protein and TolB to coordinate cell membrane integrity and virulence

Huasong Zou, Leilei Wu, Xueyan Zhu, Hang Long, Xinyu Huang, Shuying Zhu, and Qi Xue

Corresponding Author(s): Huasong Zou, Life Science and health College

Review Timeline:

Submission Date:	October 7, 2024
Editorial Decision:	November 15, 2024
Revision Received:	November 21, 2024
Accepted:	November 25, 2024

Editor: Lindsey Burbank

Reviewer(s): Disclosure of reviewer identity is with reference to reviewer comments included in decision letter(s). The following individuals involved in review of your submission have agreed to reveal their identity: Alexander N Ignatov (Reviewer #2)

Transaction Report:

DOI: <https://doi.org/10.1128/spectrum.02521-24>

Re: Spectrum02521-24 (*Xanthomonas citri* subsp. *citri* requires a genus-specific outer membrane protein and TolB to coordinate cell membrane integrity and virulence)

Dear Dr. Huasong Zou:

Thank you for the privilege of reviewing your work. Below you will find my comments, instructions from the Spectrum editorial office, and the reviewer comments.

Editor comments: Overall, the review of your manuscript found that it addresses an important plant disease, and adds to the body of knowledge around some of the genes that are important for the virulence process. The main points that should be addressed are:

1. Streamline the writing to focus more closely on the topic of the study. Some of the background discussed extensively in the introduction is not directly related to the content of the rest of the manuscript.
2. In addition, there is a lot of discussion around a mechanism for TolB-OMP regulation of the phenotypes, but not really any evidence for any specific mechanism. Instead focus on the demonstrated phenotypic changes as the main conclusion.
3. Please pay attention to the specific comments of the reviewers as outlined below.

Revision Guidelines

Sincerely,
Lindsey Burbank
Editor
Microbiology Spectrum

Reviewer #1 (Comments for the Author):

In the manuscript of Wu et al, entitled *Xanthomonas citri* sp. *citri* requires a genus specific outer membrane protein and TolB to coordinate cell membrane integrity and virulence. The authors found that omp gene mutant is deficient in stress tolerances of NaCl, sorbitol and SDS. Deletion of omp resulted in significant decrease of gum gene expression. Using pull-down assay, they found that OMP potentially interacts with TolB. In the following analyses, it revealed that tolB mutant phenocopies the deficiency as the omp mutant. This study provide valuable results to show that OMP and TolB possibly interact in cells and control bacterial stress response, virulence and EPS production. However, I have several concerns about the work, and it seems the conclusion of the manuscript is somewhat preliminary.

1. OMP and TolB are membrane bound proteins, and thus their affect on gum gene expression should be in direct. Too many unknown cascades are unidentified between these membrane proteins and downstream gene transcriptions. Therefore, most results of the work are just phenotypic change, and there is no much mechanism to explain the results of genetic analysis. After the potential binding event between TolB and OMP, what is happened?

2. Fig. 3, the authors used two round of pull-down experiment and Y2H to show that TolB and OMP interact in vitro. However, there lacks evidence from in vivo analysis so that the possibility cannot be excluded that TolB-OMP interaction is an artificial result.

3. Introduction of the manuscript lack of logic, for exmple, L55-58, why the authors suddenly mention flagellar after description of EPS, any logic connection? They described EPS, flagellar, T3SS and regulators, Tol/Pal, protease Lon and finally they studied an OMP protein. What is the topic of these pathways and descriptions? The authors should completely rewrite this part to focus on OMP.

Reviewer #2 (Comments for the Author):

The manuscript titled "*Xanthomonas citri* subsp. *citri* requires a genus-specific outer membrane protein and TolB to coordinate cell membrane integrity and virulence" is dedicated to the study of the role of *Xanthomonas*-specific outer membrane protein XAC1347 (OMP_{xan}) in the expression of pathogenicity-related genes including gum genes, TolB gene and the Type III secretion system genes, all important for *Xanthomonas* pathogenicity.

In this study, the authors found that OMP_{xan} is required for stress tolerance and cell membrane integrity, as well as the expression of gum genes for extracellular polysaccharide production. Pull-down and yeast two-hybrid assays showed that OMP_{xan} interacts with TolB, a substrate of the Tol/Pal membrane protein system. Deletion of tolB results in similar phenotypic changes as the OMP_{xan} mutant, including stress tolerance, cell membrane integrity, and expression of the type III secretion system. In contrast, the absence of TolB led to an increased level of expression of the gum genes and the production of extracellular polysaccharide. These results indicate that the interaction between OMP_{xan} and TolB coordinates multiple mechanisms to manage environmental stress and pathogenicity.

The research is dedicated to an important subject and will be of great interest to the international community. It is conducted logically and effectively using classical molecular biology techniques. The manuscript is well-written, but there are some minor corrections that need to be made.

1) The Abstract should be more specific in describing the results. Δ tolB reduces resistance to salt, not to all types of stress, and virulence is reduced by downregulating 4 hrp genes, not by regulation of all hrp genes.

2) Line 47: *Xanthomonas citri* subsp. *citri* is the causative agent of citrus canker, which causes significant commercial losses to citrus crops worldwide. However, Xcc does not occur in all citrus-growing regions, so it would be more accurate to say "in major production areas".

3) Line 48: The extracellular polysaccharides (EPS) and flagellum biosynthesis are two virulence factors that are essential for full virulence in Xcc.

Note: A) The extracellular polysaccharides (EPS) and flagellum biosynthesis are two virulence factors that are necessary for full virulence by Xcc.

However, it is possible to interpret this sentence as if they are the only factors necessary for full virulence in Xcc. Therefore, please modify the sentence to clarify this point. B) Flagellum biosynthesis was mentioned in the Introduction as an important factor for virulence, but it was not discussed in the Results or Discussion sections. Therefore, some text about the role of flagella could be removed from the Introduction.

Dear Dr. Burbank and reviewers,

Thank you for your work on our manuscript. Your kind advices are valuable in improving the quality of our manuscript. We have studied comments carefully and have made corrections which we hope meet with approval. The MS Word Track Changes function was used to revise the manuscript. Furthermore, a version without any mark was created and uploaded in submission system. The point-to-point corrections in the paper are listed below.

Sincerely

Huasong Zou

Editorial comments:

1. Streamline the writing to focus more closely on the topic of the study. Some of the background discussed extensively in the introduction is not directly related to the content of the rest of the manuscript.

Answer: Introduction was revised.

2. In addition, there is a lot of discussion around a mechanism for TolB-OMP regulation of the phenotypes, but not really any evidence for any specific mechanism. Instead focus on the demonstrated phenotypic changes as the main conclusion.

Answer: Discussion was revised.

3. Please pay attention to the specific comments of the reviewers as outlined below.

Answer: We revised the introduction and discussion focusing on the phenotypes altered in this research, according to the reviewers. In Introduction, the revision was marked with red. In discussion, deletion of the section seems no close relationship with results (previously line 239-251).

Reviewer #1 (Comments for the Author):

In the manuscript of Wu et al, entitled *Xanthomonas citri* sp. *citri* requires a genus specific outer membrane protein and TolB to coordinate cell membrane integrity and virulence. The authors found that omp gene mutant is deficient in stress tolerances of NaCl, sorbitol and SDS. Deletion of omp resulted in significant decrease of gum gene expression. Using pull-down assay, they found that OMP potentially interacts with TolB. In the following analyses, it revealed that tolB mutant phenocopies the deficiency as the omp mutant. This study provide valuable results to show that OMP and TolB possibly interact in cells and control bacterial stress response, virulence and EPS production. However, I have several concerns about the work, and it seems the conclusion of the manuscript is somewhat preliminary.

1. OMP and TolB are membrane bound proteins, and thus their affect on gum gene expression should be in direct. Too many unknown cascades are unidentified between

these membrane proteins and downstream gene transcriptions. Therefore, most results of the work are just phenotypic change, and there is no much mechanism to explain the results of genetic analysis. After the potential binding event between TolB and OMP, what is happened?

Answer: Yes, gum gene expression is regulated by several mechanisms. The mechanism why OMP or TolB affects gum gene expression has not been fully elucidated in this manuscript. We consume that the interaction between tolB and OMP help to maintain cell membrane integrity, other possible reason is that they are involved in environmental signal perceiving.

2. Fig. 3, the authors used two round of pull-down experiment and Y2H to show that TolB and OMP interact in vitro. However, there lacks evidence from in vivo analysis so that the possibility cannot be excluded that TolB-OMP interaction is an artificial result.

Answer: it is much better if can verity the interaction of TolB with OMP in vivo. Currently, we did not find a stable and reliable technique to verify this, but will try to make constructs to do bimolecular fluorescence complementation in *X. citri*.

3. Introduction of the manuscript lack of logic, for exmple, L55-58, why the authors suddenly mention flagellar after description of EPS, any logic connection? They described EPS, flagellar, T3SS and regulators, Tol/Pal, protease Lon and finally they studied an OMP protein. What is the topic of these pathways and descriptions? The authors should completely rewrite this part to focus on OMP.

Answer: The main logic idea in introduction is to explain those virulence factors of *X. citri* subsp. *citri* are affected by several circumstances. As your suggestion, the line 55-58 was corrected to focus on gum gene regulation.

Reviewer #2 (Comments for the Author):

The research is dedicated to an important subject and will be of great interest to the international community. It is conducted logically and effectively using classical molecular biology techniques. The manuscript is well-written, but there are some minor corrections that need to be made.

1) The Abstract should be more specific in describing the results. Δ tolB reduces resistance to salt, not to all types of stress, and virulence is reduced by downregulating 4 hrp genes, not by regulation of all hrp genes.

Answer: This sentence was corrected in Abstract as "The deletion of *tolB* resulted in similar phenotypic alterations as the *OMP_{Xan}* mutant in salt stress tolerance, cell membrane integrity, and the expression of *hrpG*, *hrpX*, *hrpD6*, and *hrcC* for pathogenicity. "

2) Line 47: *Xanthomonas citri* subsp. *citri* is the causative agent of citrus canker, which causes significant commercial losses to citrus crops worldwide. However, Xcc does not occur in all citrus-growing regions, so it would be more accurate to say "in major production areas".

Answer: Many thanks for this suggestion. It was corrected.

3) Line 48: The extracellular polysaccharides (EPS) and flagellum biosynthesis are two virulence factors that are essential for full virulence in Xcc.

Answer: It was corrected. The sentence and associated reference were changed.

Note: A) The extracellular polysaccharides (EPS) and flagellum biosynthesis are two virulence factors that are necessary for full virulence by Xcc.

However, it is possible to interpret this sentence as if they are the only factors necessary for full virulence in Xcc. Therefore, please modify the sentence to clarify this point. B) Flagellum biosynthesis was mentioned in the Introduction as an important factor for virulence, but it was not discussed in the Results or Discussion sections. Therefore, some text about the role of flagella could be removed from the Introduction.

Answer: Thank you for your comments. Flagellum biosynthesis mentioned in Abstract is deleted.

Re: Spectrum02521-24R1 (*Xanthomonas citri* subsp. *citri* requires a genus-specific outer membrane protein and TolB to coordinate cell membrane integrity and virulence)

Dear Dr. Huasong Zou:

Your manuscript has been accepted, and I am forwarding it to the ASM production staff for publication. Your paper will first be checked to make sure all elements meet the technical requirements. ASM staff will contact you if anything needs to be revised before copyediting and production can begin. Otherwise, you will be notified when your proofs are ready to be viewed.

Sincerely,
Lindsey Burbank
Editor
Microbiology Spectrum